# Microbial Community Response and Assembly Process of Yellow Sand Matrix in a Desert Marginal Zone Under *Morchella* Cultivation

**DOI:** 10.3390/microorganisms13040921

**Published:** 2025-04-16

**Authors:** Cuicui Su, Shengyin Zhang, Yanfang Zhou, Hao Tan, Shuncun Zhang, Tao Wang, Zhaoyun Ding, Jie Liao

**Affiliations:** 1Gansu Academy of Agri-Engineering and Technology, Lanzhou 730000, China; sucuicui963@163.com (C.S.);; 2Northwest Institute of Eco-Environment and Resource, CAS, Lanzhou 730000, China; zhangshengyin@nieer.ac.cn (S.Z.);; 3Sichuan Academy of Agricultural Sciences, Chengdu 610000, China

**Keywords:** yellow sand substrate, *Morchella*, co-occurrence network, microbial community

## Abstract

In this study, we investigated the adaptation of yellow-sand-substrate *Morchella* cultivation in the desert fringe and its effect on soil physicochemical properties and microbial communities. The qPCR and high-throughput sequencing with null modeling analyzed microbial diversity, networks, and assembly of *Morchella* cultivation under nutrient supplementation, linking physicochemical changes to microbial dynamics. The results showed that the yellow sand substrate can be planted with Morchella in the desert fringe area, as the *Morchella* cultivation with nutrient bags resulted in a yield of 691 g/m^2^ of *Morchella* fruit units. Cultivation of *Morchella* could significantly increase the physicochemical properties of the yellow sand substrate, such as soil organic matter (SOM), total nitrogen (TN), ammonium nitrogen (NH_4_^+^−N), and the microbial amount of carbon and nitrogen (MBC/MBN). The fungal community was dominated by *Ascomycota*, and *Basidiomycota*, *Firmicutes*, *Bacteroidota*, and *Actinobacteriota*. RDA analysis showed that *Ascomycota* and *Proteobacteria* were positively correlated with NH_4_^+^−N, MBN, SOM, MBC, acting potassium (AK), TN, and C/N. *Morchella* cultivation promoted a positive correlation-dominant microbial network pattern in the yellow sand substrate. The nutrient bag treatment reduced bacterial network complexity while enhancing fungal network complexity, connectivity and stability, accompanied by significant increases in *Proteobacteria*, *Bacteroidota*, *Cladosporium*, and *Thermomyces* relative abundances during cultivation until original substrate degradation. Deterministic processes dominated bacterial and fungal communities, and morel cultivation drove bacterial and fungal community assembly toward heterogeneous selection processes. The results of the study revealed the economic value of *Morchella* cultivation in the desert fringe and the application potential of improving the physicochemical properties of yellow sandy soil, which is of great importance for practical cultivation and application of morel mushrooms in the desert.

## 1. Introduction

*Morchella*, belonging to the *Ascomycota*, *Pezizomycetes*, *Pezizales*, and *Morchellaceae* families, is named for the resemblance of its fruiting bodies to a sheep’s belly. This rare soil-dwelling fungus is rich in polysaccharides, amino acids, microbiotics, minerals, proteins, and trace elements, offering benefits such as improved blood circulation, enhanced immunity, and skincare properties. Its high nutritional and medicinal value has earned it the title of “king of mushrooms” [1,2,3,4]. Due to its extremely high nutritional value and economic value, wild resources are very scarce; artificial *Morchella* cultivation has become a hot spot of research; some black lineages of wild *Morchella*, such as *M. sextelata*, *M. eximia*, and *M. importuna*, etc., have been domesticated for field soil cultivation [5,6,7]. Especially in recent years, the momentum of *Morchella* cultivation development is particularly rapid: the planting area has increased [8,9], expanding from the south to the arid desert fringe irrigated areas in the northwest, and it has become one of the main industries of rural revitalization.

Despite the rapid development of artificial *Morchella* cultivation over the past decade, challenges such as unstable production, low yields, and complete crop failure remain prevalent. In regions with limited arable land resources, *Morchella* cultivation also faces the issue of continuous cropping barriers [10]. Soil serves as the primary substrate for *Morchella* cultivation, and studies have demonstrated that its growth is closely linked to soil conditions [11]. Scholars suggest that fertile soils with high nutrient and organic matter content are conducive to higher yields, while nutrient-poor or sandy soils with low organic matter content result in lower yields. To address this, the exogenous addition of organic nutrients such as humic acid, amino acids, and fermentation broth is recommended to enhance *Morchella* production [12]. Some studies have also shown that the organic matter required for *Morchella* production of in artificial cultivation mainly comes from the carbon source provided by exogenous nutrient bags, and the soil cultivation substrate primarily provides a nitrogen source for it [13]; adding a little organic fertilizer in quartz sand substrate to cultivate *Morchella* can also achieve high yield [14].

China’s arid and semi-arid regions in the northwest are characterized by extensive deserts, including the Badanjilin, Tengger, Taklamakan, and Gurbantunggut Deserts. The ecological environment in these desert fringe areas is poor, with soils predominantly composed of nutrient-poor sandy substrates. Conventional agricultural development in these regions demands significant water and fertilizer resources, resulting in high investment costs and considerable developmental challenges [15]. However, these areas are rarely cultivated with crops, are free from soil-borne diseases and residual substances such as chemical fertilizers and pesticides, with low background levels of heavy metal content, and have a healthy soil environment [16,17]. Soil microorganisms play an important role in promoting material decomposition and transformation, nutrient cycling, and regulation of the Morita system. *Morchella* cultivation in soil substrates, desert sand substrates, and quartz sand semi-synthetic substrates significantly increased the relative abundance of dominant microbial phyla including *Proteobacteria*, *Actinobacteriota*, *Bacteroidota*, and *Ascomycota*. This cultivation practice also enhanced certain physicochemical characteristics, such as organic matter content and total nitrogen levels, in these substrates [13,18,19]. In ecosystems, individual microorganisms do not exist in isolation, but form co-occurring networks by interacting directly or indirectly with each other, and the response of microbial communities to the environment is reflected by the complexity and stability of the network [20,21]. In responding to the environment, the microbial community assembles through deterministic and stochastic processes differently [22]. It was found that an exogenous supply of organic matter can promote the microbial assembly process from stochastic to deterministic, while increased changes in soil nutrients, among others, can also lead to increased heterogeneity of the soil environment [23]. The organic matter and nitrogen content of wind-sand soils in the desert fringe area are very low, and there are fewer studies on the adaptability of *Morchella* cultivation in this area, as well as the potential and mechanism of soil improvement. Based on this, we investigated the variations in the physicochemical properties, microbial community structure, and assembly process of the yellow sand substrate under *Morchella* cultivation with nutrient bags and *Morchella* cultivation without nutrient bags. In this study, we analyzed the mechanism of the synergistic effect of the *Morchella*-microbial-soil system during the cultivation of morel mushrooms in the desert fringe area by qPCR and high-throughput sequencing technology. This study provides a theoretical reference for the development of the desert *Morchella* industry.

## 2. Materials and Methods

### 2.1. Overview of the Experimental Area

This experiment was conducted in the Gobi Agricultural Demonstration Base of Wudaogou, Huanghuatan, Gulang County, Wuwei City, Gansu Province in China (latitude 36.04 N, longitude 103.86 E, elevation 1520 m), which is located at the southern edge of the Tengger Desert. The wind-sand soil, which had not been planted with *Morchella* and had not been fertilized with chemical and organic fertilizers, was selected for *Morchella* cultivation. The SOM content of the wind-borne sand soil in the test area was 1.27 g/kg, the TN was 0.080 g/kg, the effective phosphorus (AP) content was 70.45 mg/kg, the AK content was 111 mg/kg, the pH value was 8.83, and the particle size composition was as follows: the proportion of particles ≤0. 075 mm was zero, the proportion of particles in the range of 0.075–0.20 mm was 68.7%, the proportion of particles in the range of 0.20–1.00 mm was 31.7%, and the proportion of particles in the range of 0.20–1.00 mm was 31.6%.

### 2.2. Materials

The variety of morel mushroom for the test was the six sister morel mushroom (*Morchella sextelata*) “Sichuan morel mushroom No. 6′′. The nutrition bag formula for wheat grain and hulls was in accordance with the mass fraction of 4:1 ratio with the mix, produced by Sichuan Jindi Tianlingjian Biotechnology Co. (Chengdu, China).

### 2.3. Methods

#### 2.3.1. Experimental Design

The experiment was designed with three groups of treatments: yellow sand substrate without *Morchella* cultivation (YSC), yellow sand substrate of *Morchella* cultivation without nutrient bags (YSM), and yellow sand substrate of *Morchella* cultivation and 10 nutrient bags per square meter (YSMN); each treatment was composed of five replicated blocks, each with a 2.0 m in length and 1.0 m in width. Sampling in the sandy soil substrate was carried out before cultivation, during mushroom emergence after cultivation, and after harvest, with the day of sowing as the starting point of sampling, and 1 sample was taken every 30 days, and a total of 5 samples were taken in the 0th (A), 30th (B), 60th (C), 90th (D), 120th (E), and 150th (F) days after sowing of *Morchella* mushrooms. Among them, the mushroom emergence and harvesting stages occurred at the 60 d to 90 d stages after sowing morel mushrooms. Soil samples collected at these six time points (A–F) were analyzed for soil physicochemical properties, and soil samples collected on the day of sowing (time point A, 0 d after sowing) and 2 months after harvesting (time point F, 150 d after sowing) were analyzed for microbial community structure. In the control and treatment groups, 75 g of quicklime (Wuwei Hongyuan Municipal Engineering Co., Wuwei, China), 30 g of K_2_NO_3_ (Maoming Xiongda Chemical Co., Maoming, China), 60 g of CaSO_4_ (Sinopharm Chemical Reagent Co., Shanghai, China), and 10 g of K_2_HPO_4_ (Tianjin Comeo Chemical Reagent Co., Tianjin, China) per square meter were applied prior to the *Morchella* cultivation.

#### 2.3.2. Sample Collection

In each plot, five sampling points were selected outside the nutrient bag positions according to the “S”-shaped sampling method. Sandy substrate samples were collected using a soil auger with a diameter of 0.5 cm and a length of 20 cm, then thoroughly mixed and divided into three parts. One portion was frozen by liquid nitrogen and then stored in the refrigerator at −80 °C for the determination of soil microorganisms. One part was frozen with dry ice to bring back to the laboratory. One copy was stored in the refrigerator at −20 °C for the determination of MBC, MBN, NH_4_^+^−N, and NO_3_^−^−N, and one copy was naturally dried for the determination of basic physical and chemical properties.

#### 2.3.3. Determination of Physical and Chemical Properties

The physicochemical properties of the yellow sand substrate soil were determined with reference to soil agrochemical analysis [24,25]. Soil organic matter (SOM) was oxidized by potassium dichromate; total nitrogen (TN) was determined by the automatic Kjeldahl method; active phosphorus (AP) was extracted by the sodium bicarbonate leaching–molybdenum–antimony colorimetric method; active potassium (AK) was extracted by the ammonium acetate leaching–flame photometric method; ammonium nitrogen (NH_4_^+^−N) was extracted by the potassium chloride leaching–indophenol blue colorimetric method; nitrate nitrogen (NO_3_^−^N) was extracted by the potassium chloride leaching–double wave colorimetric method; microbial biomass carbon and nitrogen (MBC/MBN) were determined by chloroform fumigation–K_2_SO_4_ leaching and a TOC analyzer.

#### 2.3.4. Extraction and Sequencing of Soil Microbial DNA

The yellow sand substrate samples were collected, the total DNA of the samples was extracted using the Magnetic Soil and Stool DNA Kit TS001 (Tiangen Biochemical Technology (Beijing, China) Co., Ltd., Model: DP 712), and the purity and concentration of DNA were determined by agarose gel electrophoresis. The Magnetic Soil and Stool DNA Kit TS001 Kit ddH_2_O was also used as a blank control. The bacterial amplified region was the V3 + V4 variable region of the 16S rRNA gene, and the universal primers were 338 F: ACTCCTACGGGAGGCAGCA, 806 R: GGACTACHVGGGTWTCTAAT; the fungal amplified region was ITS2 and the universal primers were ITS1 F: CTTGGTCATTTAGAGGAAGTAA ITS2:GCTGCGTTCTTCATCGATGC; PCR amplification system: Genomic DNA 2.5–4.0 ng, Vn F (10 μmol/L) 0.3 μL, Vn R (10 μmol/L) 0.3 μL, KOD FX Neo buffer (Beijing Bailinke Biotechnology Co., Beijing, China) 5 μL, and dNTP (2 mmol/L) 2 μL. KOD FX Neo (Beijing Bailinke Biotechnology Co., Beijing, China) 0.2 μL, and ddH_2_O made up to 10 μL. PCR reaction conditions: 95 °C for 5 min; 95 °C for 30 s, 5 °C for 30 s, 72 °C for 40 s, 20 cycles; 72 °C for 7 min. The amplified products were purified with Omega DNA purification kit (Omega Inc., Norcross, GA, USA) and quantified using Qsep-400 (BiOptic, Inc., New Taipei City, Taiwan). The amplicon library was paired-end sequenced (2 × 250) on an Illumina Novaseq 6000 (Beijing Biomarker Technologies Co., Ltd., Beijing, China).

#### 2.3.5. Real-Time Quantitative PCR (qPCR) Analysis [26]

Soil DNA was extracted using Bio-Base-X Pure Soil DNA Extraction Kit (Shandong Boco Bio-industry Co., Jinan, China, Item No. M2022). The extracted DNA was separated by 10 g/L agarose gel electrophoresis, amplified and plasmids constructed with primers corresponding to fungi and bacteria in Section 2.3.4, and detected by OD using an ultramicro nucleic acid protein assay (scandrop100) using the A260/A280 ratio. Real-time quantitative PCR (qPCR) analysis was performed using an analytic jena-qTOWER2.2 fluorescence quantitative PCR instrument (Jena, Germany). qPCR amplification of each sample was set up with three technical replicates, while three negative controls were set up to monitor the degree of contamination of the samples. qPCR reaction program: 94 °C for 4 min, 1 cycle; 94 °C for 20 s, 63 °C for 30 s, 72 °C 30 s + plate read + plate read, 39 cycles; melt curve analysis, 60 °C~95 °C, +1 °C/cycle, hold time 4 s. The qPCR amplification system was 2 × NovoStart^®^SYBR^®^Green Super mix (Beijing Adelaide Biotechnology Co., Beijing, China) 5 μL, forward primer (200 nmol/L) 0.5 μL, reverse primer (200 nmol/L) 0.5 μL, 10× diluted soil DNA sample 1 μL, ddH_2_O_3_ μL. The standard curve was generated by gradient dilution of monoclonal plasmids containing amplified genes, ITS plasmid samples were diluted according to 108~103, 16S plasmid samples were diluted according to 10^7^~10^10^ dilution, each reaction was 3 replicates, and the amplification results showed that the negative control was free of contamination. A standard curve was generated by plotting the logarithmic values of standard concentrations (x-axis) against corresponding Ct values (y-axis). The copy numbers of ITS and bacterial 16S genes in the sandy soil matrix were subsequently quantified using the external standard method.

#### 2.3.6. Bioinformatic Analysis

Raw sequencing data were sequenced by constructing small fragment libraries using paired-end sequencing, and high-throughput sequencing of soil samples was performed on the Illumina NovaSeq 6000 platform. The raw image data files obtained from high-throughput sequencing were converted into sequenced reads by base calling analysis, and the raw reads obtained from sequencing were filtered using Trimmoactic Version 1.2.v0.33 software. Then primer reads were processed using cutadapt 1.9.1 software for primer sequence identification and removal, followed by USEARCH 10.0 for bipartite read splicing and chimera removal (UCHIME 8.1), and finally, high-quality sequences (non-chimeric reads) were obtained for subsequent analysis. The qualified sequences with more than 97% similarity thresholds were allocated to one operational taxonomic unit (OTU) using USEARCH (version 10.0). Clean reads then were conducted on feature classification to output an ASVs (amplicon sequence variants) by DADA2, and the ASVs conuts less than 2 in all samples were filtered. The QIIME2 2020.6 software used the UNITE (fungi) and Silva 138 (bacteria) reference databases to taxonomically annotate feature sequences, employing a naive Bayesian classifier combined with contrasting classifiers, thereby obtaining species-level taxonomic information for each feature. The community composition of each sample was then analyzed at multiple taxonomic levels (phylum, class, order, family, genus, and species) and visualized using “R” tools (version 4.3.2). The LEfSe model (LDA > 4.0) was used to analyze the signature taxa (https://www.biocloud.net).

Using high-throughput sequencing, representative operational units (OTUs) were selected, phylogenetic ecological networks were constructed using RMT, and MENAP was used for network construction and determination of network attribute parameters, and the resulting basic network attribute files were used for network visualization using Gephi 0.1.1 software [27]. The similarity of soil microbial communities was calculated using the Bray–Curits distance with the “Vegan” package of R. Based on the “Picante” package of R, a null model was performed. A null model was performed to analyze the βNTI (βnearest taxon index) and the Raup–Crick (RCbray) index of Bray–Curits to determine the proportion of contribution of deterministic and stochastic processes in community assembly [28]. Neutral model analysis was performed using the “Stats4” (version 3.6.2) and “Hmisc” (version 5.2-3) packages in R to determine the influence of stochastic processes on community assembly.

### 2.4. Data Analysis and Statistics

Data were processed using Excel, analyzed by ANOVA for data significance using SPSS 21.0, and plotted using Origin 24.0.

## 3. Results

### 3.1. Yield of Morchella

The YSM did not yield fresh *Morchella* fruiting bodies, whereas the YSMN achieved a yield of 691 ± 23.72 g/m^2^. This indicates that *Morchella* can be cultivated in yellow sandy soil at the desert fringe, and that the exogenous addition of nutrient bags significantly increased yield.

### 3.2. Physicochemical Properties of the Yellow Sand Substrate

The yellow sand substrate for *Morchella* cultivation was weakly alkaline, with a pH of about 8.84. The pH value of YSM increased and then decreased between 30 and 90 d after sowing, and the pH value of YSMN decreased and then increased, and there was no significant change in the pH value of the yellow sand substrate before cultivation and after harvesting *Morchella* (Figure 1) The background organic matter of the yellow sand substrate was 1.27 g/kg. After sowing *Morchella*, the SOM content had no significant change in the YSC group, and reached a maximum of 2.67 g/kg 60 d after sowing in the YSM group. In the YSMN group, the SOM content increased and then decreased with the time after sowing; it reached 4.54 g/kg 30 d after sowing, and then stabilized at 120~150 d after harvesting *Morchella*, but it was still higher than the initial background value. The SOM content of the YSM and YSMN groups increased by 46.7% and 3.4 times, respectively, compared with the YMC group (*p* < 0.01).

After the *Morchella* cultivation, the TN content and NH_4_^+^−N content in the yellow sand substrate showed an overall increasing trend at the mushroom stage, in which the NH_4_^+^−N content in the YSM group decreased at 30 d after sowing and then increased significantly from 60 to 90 d (*p* < 0.01). The TN content in the YSM and YSMN groups increased by 75.56% and 36.09%, respectively, compared with that of the YSC group. The NH_4_^+^−N content increased by 39.46% and 52.06%, respectively, compared with that of the YSC group (*p* < 0.05). There was no significant change of TN and NH_4_^+^−N in the whole process of YSC in the YSC group. The NO_3_^−^N content of the yellow sand substrate with the prolongation of time after sowing showed a general trend of decline in the treatment, in which the YSMN group 30 d after sowing showed the largest decline of 79.76%, the YSM group 90 d after sowing group the largest decline of 88.13%, and then the change became more moderate; the YSC group first decreased and then increased and then decreased. After the end of harvest, the NO_3_^-^N content of the YSM and YSMN groups was significantly lower than that of the YSC group (*p* < 0.01). The AK content of the yellow sand substrate in the YSM and YSMN groups decreased and then increased (*p* < 0.01), with a minimum of 63 mg/kg in the YSM group at 60 d after sowing, and a minimum of 57 mg/kg in the YSMN group at 30 d after sowing, and then both groups returned to a similar level as that of the YSC group 150 d after sowing. The AP content increased and then decreased from 0 to 150 d after sowing, reaching a maximum at 30 d after sowing in the YSM group and 90 d after sowing in the YSMN group, and increasing by 25.19% and 13.39%, respectively, compared to the YSC group 150 d after sowing (*p* < 0.05).

From 0 to 150 d after sowing of *Morchella*, there was no significant change in the MBC content of the yellow sand substrate in the YSC group. The MBC content of The YSMN group gradually increased and was significantly higher than YSC and YSM groups 60 d after sowing (*p* < 0. 05). The YSM group gradually increased after the first after sowing, and was higher than the YSC group 90 d after sowing (*p* < 0.05). At 150 d after sowing, the YSM and YSMN groups were 1.62 times and 3.04 times higher than that of the YSC group (*p* < 0.01). The MBN content of the YSC group suddenly decreased 90 d after sowing. The MBN content of the YSM group first decreased and then increased, and, 150 d after sowing, was equal to the initial level (*p* < 0.01). The MBN content of the YSMN group first increased and then decreased from sowing to the mushroom stage, then increased again after the end of harvest, and increased by 93.80% compared with the initial level (*p* < 0.01). The C/N of the YSM group gradually increased after decreasing 30 d after sowing, and it kept increasing throughout the stage in the YSMN group; the YSC increased after no significant change from 0 to 120 d. The C/N of the YSM and YSMN groups 150 d after sowing were 72% and 1.9 times higher than the YSC group, respectively (*p* < 0.01).

### 3.3. Bacterial 16S and Fungal ITS Content

Bacterial 16S and fungal ITS plasmid samples were amplified by qPCR technology, and the bands were clear without trailing; the amplification curve and lysis curve were reproducible. The amplification curve was a standard “S” curve, and the lysis curve was a single peak (Figure 2), indicating that the ITS and 16S plasmids had good reproducibility and specificity and met the requirements for subsequent analysis. The amplification R^2^ of the 16S plasmid was 0.996, with an amplification efficiency of 0.88, and the amplification R^2^ of ITS plasmid was 0.999, with an amplification efficiency of 1.01, which was able to detect the bacterial and fungal contents of the yellow sand substrate samples very well. The copy numbers of ITS and 16S genes in the yellow sand substrate of cultivated and uncultivated *Morchella* were determined by standard curves (Figure 3). The *Morchella* cultivation had a significant effect on the fungal and bacterial content of the yellow sand substrate soil. There was no significant difference in the fungal and bacterial copy numbers of the YSC between the day of sowing (A) and the 150 days after sowing (F), but the fungal and bacterial copy numbers of the YSM and YSMN groups increased significantly (*p* < 0.05), the fungal copy number of YSM increased from 2.09 × 10^2^ to 2.15 × 10^3^, and the bacterial copy number increased from 1.43 × 10^2^ to 1.40 × 10^3^, while the fungal copy number of the YSMN increased from 2.72 × 10^2^ to 2.74 × 10^3^. The fungal copy numbers of the YSMN and YSM showed no significant difference, but were significantly higher than that of the YSC at 150 days after sowing (*p* < 0.05), while the bacterial copy numbers of the three treatments differed significantly from each other (*p* < 0.05).

### 3.4. Microbial Community Diversity Analysis

Bacterial and fungal DNA of yellow sand substrate from different treatments were extracted and analyzed by high-throughput sequencing. A total of 78,435 OTUs of bacterial species were obtained with more than 97% similarity, belonging to 40 phyla and 926 genera, with a total of 1,444,158 sequences and an average of 38,897 to 61,148 sequences. A total of 22,813 OTUs of fungal species were obtained with more than 97% similarity, belonging to 19 phyla and 1168 genera, and an average of 38,417 to 65,950 sequences for each sample. The yellow sand substrate of the YSC, YSM, and YSMN groups shared 1541 bacterial OTUs and 1181 fungal OTUs; the numbers of matrix-specific bacterial OTUs and fungal OTUs of the three groups were 22,386, 26,568, and 24,617 (Figure 4a), respectively, and the numbers of fungal OTUs were 6346, 7760, 7660, and 7660 (Figure 4b), respectively. The total proportion of the fungal community was higher than that of the bacterial community, indicating that the succession of species composition of the bacterial community in the yellow sand substrate after *Morchella* cultivation was greater than that of the fungal community. At time point F, the Shannon index of the bacterial community was lower than the initial level (A), in which the Shannon index of the YSM and YSMN groups was slightly higher than the YSC group. The diversity of the fungal community was also lower than that of the initial state, and the Shannon index of the YSM group was higher than that of the YSC group (*p* < 0.05), and that of the YSMN group was lower than that of the YSC group, and the difference between the Shannon index of the YSM group and that of the YSMN group was significant (Figure 5).

The NMDS analysis showed that the bacterial and fungal communities in the yellow sand substrate of *Morchella* cultivation after two months did not differ significantly between treatments (Figure 6), but the distribution pattern of the points within the same treatment revealed the local effects of *Morchella* cultivation on the microbial communities in the yellow sand substrate. The distribution of points in the YSMN group was more concentrated, suggesting that the addition of the nutrient bags may have promoted the homogenization of the microbial communities through the provision of exogenous carbon sources, and the composition of the microbial communities tended to be consistent. The dispersed points in the YSM group may be related to the competitive use of local nutrients by morelloomycetes in the absence of nutrient bags, while the points in the YSC control group were more widely distributed, reflecting the natural heterogeneity of bacterial communities in the undisturbed yellow sand substrate. It can be seen that the addition of exogenous nutrient bags for *Morchella* cultivation contributed to the differentiation of the microbial community composition of the yellow sand mechanism at the micro-scale.

### 3.5. Microbial Community Compositional Variation Analysis

The dominant phyla of the bacterial community in the yellow sand substrate at 0 days (A) and 150 days (F) after sowing of both cultivated and uncultivated *Morchella* treatments were dominated by *Proteobacteria*, *Firmicutes*, *Bacteroidota*, and *Actinobacteriota* (Figure 7a). The relative abundance of each bacterial community did not change much on the tine-point A and F in the YSC group, with *Firmicutes* accounting for the largest proportion of 27.59%, followed by *Bacteroidota* with 21.20%, *Proteobacteria* with 20.39%, and *Actinobacteriota* with 10.77%. Compared with the initial level in the YSM group, except for *Actinobacteriota*, the other three dominant phyla showed an increase, of which the proportion of *Proteobacteria* was the largest at 22.75%, followed by *Firmicutes* at 22.15%, *Bacteroidota* at 21.24%, and *Actinobacteriota* at 9.61%. In the YSMN group, the proportion of *Proteobacteria* was significantly increased, with the largest proportion of 30.71%, followed by 18.88% for *Firmicutes*, 17.96% for *Bacteroidota*, and 10.26% for *Actinobacteriota*. At the genus level, unclassified_Muribaculaceae and unclassified_Bacteria showed a reduction in both YSM and YSMN substrates of *Morchella* cultivation (Figure 7b).

The dominant phyla of the fungal communities were both *Ascomycota* and *Basidiomycota* (Figure 7c), and the dominant phyla ranked No. 1 in each sample were all *Ascomycota*, which is also the phylum to which the *Morchella* itself belongs. Compared with the initial state (A), the proportion of *Ascomyco*ta in the YSC and the YSM groups at time point F decreased to 75.09% and 74.44%, and the proportion of the *Basidiomycota* increased to 12.04% and 15.66%, respectively. The percentage of *Ascomycota* in the YSMN group was higher than that of the initial state and significantly higher than that of the YSC and YSM groups; the percentage of *Basidiomycota* was lower and significantly lower than that of the YSC and YSM group. At the genus Genus level, compared to time point A, time point F showed a decrease in the relative abundance of the YSM *Aspergillus*, an increase in the relative abundance of *Cladosporium* in YSMN than in YSM, a decrease in the relative abundance of *Thermomyces* in YSC, no significant change in YSM, and an increase in the performance of YSMN, which was significantly higher than that of both YSC and YSM (Figure 7d).

LEfSe analysis of the dominant phyla of bacteria and fungi in the yellow sand substrate of the *Morchella* cultivation for time points A and F (Figure 8) showed that the biomarker taxa of bacteria in the yellow sand substrate of the *Morchella* cultivation were *AlphaProteobactieria* and *Proteobacteria*, and the biomarker taxa of fungi were *Sordariomycetes*, which reflected the dominant microbial species established by the *Morchella* cultivation in the yellow sand substrate.

### 3.6. Correlation Analysis Between the Microbial Community and Environmental Factors of Yellow Sand Substrate

Taking the top 4 bacterial dominant phyla in terms of relative abundance, the degree of influence of the environmental factors of the yellow sand substrate on them was AP > NH_4_^+^N > TN > AK > MBN > pH > SOM > NO_3_^−^−N > C/N > MBC (Figure 9a) *Proteobacteria* were positively correlated with NH_4_^+^N, MBN, SOM, MBC, AK, TN, and C/N, and negatively correlated with pH, AP, and NO_3_^−^−N, and also negatively correlated with NH_4_^+^−N, SOM, and C/N, and positively and significantly correlated with NH_4_^+^−N, SOM, and MBN. *Actinobacteriota* was positively correlated with NH_4_^+^−N, MBN, SOM, MBC, TN, MBC, AK, and NO_3_^−^−N, negatively correlated with AP, pH, and C/N, and positively and significantly correlated with NH_4_^+^−N, and TN. *Firmicutes* was positively correlated with C/N, AK, pH, AP, and negatively correlated with other environmental factors. *Bacteroidota* was positively and significantly correlated with AK, MBC, SOM, MBN, and MBC, and negatively correlated with other environmental factors. *Ascomycota* and *Basidiomycota*, whose relative abundance accounted for a relatively large proportion of the total, was selected, and the influence of the environmental factors of the yellow sand substrate on them was NH_4_^+^−N > MBN > TN > SOM > MBC > AP > CN > AK > pH > NO_3_^−^−N (Figure 9b). *Ascomycota* was negatively correlated with NH_4_^+^−N, MBN, MBC, SOM, C/N, TN, pH, AK, NO_3_^−^−N, and negatively correlated with AP. *Basidiomycota* was positively correlated with TN, AP, AK, and NO_3_^−^−N, and negatively correlated with other environmental factors.

### 3.7. Microbial Community Correlation Network Analysis

In order to clarify the microbial co-occurrence pattern of yellow sand substrate under the planting of *Morchella*, the network diagrams of bacterial and fungal co-occurrence based on the level of OTUs were constructed, respectively (Figure 10 and Figure 11). The results showed (Table 1) that the number of nodes, connections, average degree of nodes, average clustering coefficient, and average path distance of the YSM and YSMN bacterial networks were decreased, and the network transmissibility and network density were increased compared with YSC, indicating that the *Morchella* cultivation reduced the complexity of bacterial networks, but the connectivity of the networks was better, in which the number of positive connections of the YSMN group was increased by 31. 6%, which indicated that *Morchella* cultivation bag material promoted the bacterial network pattern of the yellow sand substrate to be dominated by positive correlation patterns. The positive connection number of the fungal network of YSC, YSM, and YSMN were 66.7%, 54.9%, and 87.4%, respectively, which indicated that the fungal network patterns of the yellow sand substrate had all been dominated by positive correlation patterns. Compared with YSC, the fungal co-occurrence network in YSM exhibited a reduction in node numbers, link quantities, and positively correlated connections, suggesting that Morchella cultivation without bag substrates reduced fungal competitive interactions and decreased network complexity. The average node degree, average clustering coefficient, average path distance, and network transmissibility and network density increased in the YSM, indicating that the stability of the fungal network increased. In the YSMN fungal network, the node number, connection count, and positive correlation connections increased, while the node average degree, average clustering coefficient, and average path distance decreased, accompanied by increases in network transmissibility and density. These topological changes demonstrate that *Morchella* cultivation with nutrient bags enhanced fungal network complexity, thereby improving network connectivity and stability while strengthening resistance to external environmental perturbations.

### 3.8. Microbial Community Assembly Processes Based on iCAMP Modeling

Deterministic processes include homogeneous and heterogeneous selection, and stochastic processes include homogeneous diffusion, diffusion limitation, and uncertainty processes. In the cultivation of *Morchella*, deterministic processes dominated the assembly of bacteria and fungi (Table 2), especially in the YSMN group. Compared to YSC, the YSM bacterial and fungal community assembly processes did not change significantly, while the YSMN treatment increased bacterial deterministic processes by 30.12% and fungal deterministic processes by 31.11%. Although deterministic processes dominated the bacterial assembly and fungal community assembly of YSMN, the deterministic processes had different compositions (Table 3). Homogeneous diffusion of YSC and YSM explained 55.56% of the fungal community assembly in the yellow sand substrate, and heterogeneous selection of YSMN explained 46.67% of the fungal community assembly in soil. Heterogeneous selection dominated the process of microbial community assembly in yellow sandy substrate soil by *Morchella* cultivation, and homogeneous diffusion, diffusion limitation, and non-dominant processes had less effect on bacterial assembly.

## 4. Discussion

The results of this study demonstrated that *Morchella* cultivation in yellow sand substrates at the desert fringe, with the addition of nutrient bags, achieved a fruiting body yield of 691 g/m^2^. Additionally, the soil organic matter (SOM) and nitrogen content increased significantly. However, the SOM content in the yellow sand substrate remained considerably lower than that of other soil substrates and quartz sand semi-synthetic substrates [14,29]. These findings demonstrate that alkaline sandy soils at desert margins represent a potential cultivation substrate for Morchella with both ecological and economic value. Furthermore, our results indicate no significant correlation between Morchella yield and substrate organic matter content, suggesting that high-organic-matter soils are not an essential prerequisite for successful *Morchella* cultivation [30]. *Morchella* cultivation had the greatest impact on the physicochemical properties of the yellow sand substrate by significantly increasing the SOM content, to 3.4 times higher compared to YSC (*p* < 0.01). This observed increase may be attributed to the extension of Morchella mycelia from the cultivation substrate into adjacent nutrient bags, facilitating the decomposition of carbon-rich organic matter and subsequent translocation of assimilated nutrients back to the yellow sand matrix to support fungal biomass accumulation. Consistent with findings by Tan Hao [30], Liu X. [31], we also observed a significant increase rise in the C/N of the YSMN group, further supporting the role of nutrient redistribution in modifying substrate properties. The TN and NH_4_^+^−N contents of the yellow sand substrate showed an increasing trend during the growth stage of *Morchella*, which, in the YSMN group, were still significantly higher than in the YSC group two months after the end of the substrate harvesting (*p* < 0.01), while the NO_3_^−^−N content exhibited an opposite trend. These results indicated that exogenous nutrient bags provided nutrient support for *Morchella* growth, and altered the soil nitrogen cycling pattern, which aligns with the findings from Zhang Y, et al. [32] in Shaanxi. The AP content in the yellow sand substrate firstly increased in the mushroom stage; the AK content decreased, and returned to near the initial level after the substrate was harvested, which was slightly different from the conclusion of the study of Zhang, Y. et al. [18] in the substrate of in the soil, but this might be due to the differences in the ecological environments.

The addition of exogenous nutrient bags for *Morchella* cultivation not only improved the physicochemical properties of the yellow sand substrate and changed the soil nutrient cycling pattern, but also provided additional resource support for microbial community interactions. In this study, the bacterial Shannon index in the YSM and YSMN groups was slightly higher than that in the YSC, while fungal diversity showed a more pronounced reduction in the YSMN group alongside significant increase in the fungal copies number (*p* < 0.01). These results suggest that exogenous nutrient bags exacerbated the competition of fungal communities in the *Morchella* cultivation substrate, which is consistent with the conclusions of the study by Yue Y. et al. [27]. The observed reduction in background microbial diversity may be attributed to the vigorous mycelial proliferation of *Morchella*, which became the dominant fungal community in the nutrient-amended yellow sand substrate. This phenomenon aligns with the findings of Zhang et al. [32], who similarly reported decreased microbial diversity in *Morchella* cultivation soils under non-contiguous cropping (NCC) conditions. After the *Morchella* cultivation, the dominant fungal phylum *Ascomycota* tended to be the most optimally dominant group, and the amplification and enrichment of those placed in YSMN was more robust than in YSM. This is in agreement with the dominant flora of the inter-root soils of *Morchella* in Shaanxi, China, and in Neuchâtel, Switzerland [18,33], confirming the phenomenon that the organic matter, microbial mass carbon, and nitrogen of the yellow sand substrate were greatly enhanced after *Morchella* cultivation [34,35,36]. The relative abundance of fungal genera that promote organic matter decomposition and nutrient cycling, such as *Cladosporium* and *Thermomyces*, remained high in relative abundance percentage after the degradation of the original base of the *Morchella*, which also explains the changes in soil physicochemical properties. The dominant bacterial phyla were mainly *Firmicutes*, *Proteobacteria*, *Bacteroidota,* and *Actinobacteriota*, in agreement with the findings of Shen C. [37], Moretti G.L. [38]. *Morchella* cultivation resulted in an increase in the relative abundance of the *Proteobacteria* and a decrease in the relative abundance of *Firmicutes*, which was consistent with the increase in SOM, NH_4_^+^−N, and TN content. RDA analysis also showed that *Ascomycota* was positively correlated with TN, NH_4_^+^−N, SOM, MBC, and MBN, and *Proteobacteria* was positively correlated with TN, AP, and NH_4_^+^−N, and negatively correlated with AK. Currently reported research results [11,39,40,41] similarly found that soil environmental factors such as the NH_4_^+^−N content, SOM content, AP content, AK content, and other soil environmental factors on the growth of *Morchella* significantly. Notably, studies specific to the Jilin region [42] have identified TN, SOM, and AP as the most critical soil factors for the dominant fungal colonization of *Morchella* rhizosphere.

Microbial ecological networks serve as a critical tool for studying dynamic interspecies relationships. Co-occurrence network analysis can elucidate the assembly mechanisms and interactions within soil microbial communities, offering valuable insights into ecosystem complexity and stability [43,44]. In this study, we found that there was a significant difference between bacterial and fungal communities in response to *Morchella* cultivation: the number of positive connections of bacterial network in the YSM group increased, while the number of positive connections of fungal network decreased, whereas the number of positive connections of both bacterial and fungal networks in the YSMN group increased dramatically. This result suggests that *Morchella* cultivation enhanced the symbiosis and interactions among microbial communities, which may be related to the additional nutrients provided by the exogenous nutrient bags. These supplemental nutrients likely influenced metabolic patterns, thereby influencing both community structure and interspecies interactions. The symbiotic network connecting lines and the average path length were correlated with the network structure: the higher the number of network connecting lines, the more complex the network structure, and the smaller the average path, the smaller the distance between its nodes and the tighter the network connecting structure [20]. The bacterial symbiotic network in the YSMN group exhibited significantly reduced network complexity relative to the YSC group, as evidenced by decreased connection numbers, lower average node degree, and shorter average path distance. Notably, despite this simplified network architecture, the YSMN group demonstrated enhanced network connectivity, suggesting more efficient inter-node interactions. In the fungal symbiotic network, the connection number, node average degree, average path distance, and transmissibility increased, network complexity increased, and the network connectivity structure was more tightly connected. These results suggest that the *Morchella* cultivation significantly altered the topology of bacterial and fungal symbiotic networks in the yellow sand substrate, especially in the YSMN, which resulted in a more sensitive response of the microbial community and a more complex and stable structure of the symbiotic network, thus improving the ecosystem function. This result is consistent with the findings of Morrien E et al. [45] and further supports the idea that *Morchella* cultivation can increase the number of symbiotic network nodes [26]. In addition, microbial community assembly processes are primarily driven by deterministic processes (e.g., heterogeneous selection) and stochastic processes (e.g., homogeneous diffusion). In this study, deterministic processes dominated bacterial and fungal community assembly, especially in the YSMN group, and the relative importance of heterogeneous selection increased significantly. This is consistent with previous findings that changes in SOM enhance heterogeneous selection, leading to changes in microbial community assembly patterns [46,47]. Stronger heterogeneous selectivity suggests that the yellow sand substrate for *Morchella* cultivation can accommodate more microorganisms with different resource utilization strategies to coexist, and these microorganisms promote soil nutrient cycling and utilization through interactions [24]. The study by Yue Y H et al. [27] also showed that the dynamic changes in the soil microbial community were closely related to the fluctuation of soil physicochemical properties, which further explains that in the present study, the nutrient content of the YSMN group was significantly higher than that of the YSC group.

## 5. Conclusions

*Morchella* cultivation in yellow sandy soil at the desert fringe, with the addition of nutrient bags, resulted in a fruiting body yield of 691 g/m^2^. The soil organic matter (SOM), total nitrogen (TN), and ammonium nitrogen (NH_4_^+^−N) contents of the yellow sand substrate increased significantly during the growing period, with the SOM content increasing by 3.4-fold in the nutrient bag treatment. The *Morchella* cultivation significantly changed the microbial community structure of the yellow sand substrate; the succession of the bacterial community was greater than that of the fungal community; *Ascomycota* and *Proteobacteria* became the dominant bacterial groups; and the relative abundance of *Ascomycota* was significantly increased in the treatment of placing nutrient bags. The *Morchella* cultivation affected the structure of the co-occurrence network of microorganisms and enhanced the deterministic process of microbial community in the yellow sand substrate; the contribution of heterogeneous selection in microbial assembly was especially significantly increased. This study confirmed the feasibility of yellow sandy soil as a potential substrate for *Morchella* cultivation in the desert fringe zone, revealed the key role of exogenous nutrient bags for *Morchella* cultivation in enhancing soil nutrients and microbial interactions, and provided a new idea for high value-added utilization as well as soil improvement and ecological restoration of the desert fringe zone.

## Figures and Tables

**Figure 1 microorganisms-13-00921-f001:**
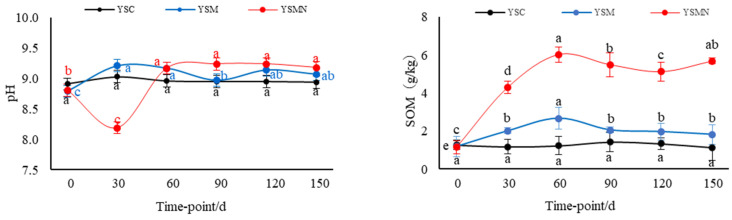
Shift of physiochemical chracteristics in the yellow sand substrate during *Morchella* cultivation. Note: SOM stands for soil organic matter, TN stands for total nitrogen, NH_4_^+^−N stands for ammonium nitrogen, NO_3_^−^−N stands for nitrate nitrogen, AK stands for available potassium, AP stands for available phosphorus, MBC stands for microbial biomass carbon, MBN stands for microbial biomass nitrogen, and C/N stands for carbon/nitrogen ratio. YSC stands for yellow sand substrate without *Morchella* cultivation, YSM stands for yellow sand substrate *Morchella* cultivation without nutrient bags, YSMN stands for yellow sand substrate *Morchella* cultivation with nutrient bags. Different lowercase letters indicate significant differences (*p* < 0.05) in the same treatment under different sampling time points, where black represents YSC treatment, blue represents YSM treatment, and red represents YSMN treatment.

**Figure 2 microorganisms-13-00921-f002:**
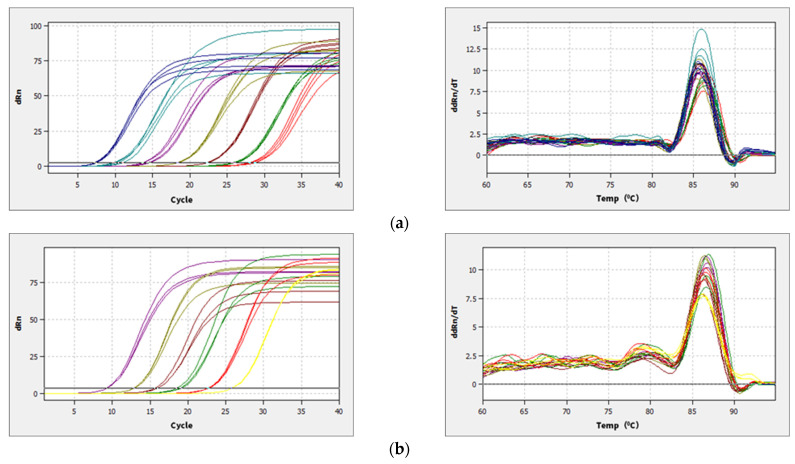
qPCR amplification curves (**left**) and melting curves (**right**) of plasmid samples of gene 16S (**a**) and ITS (**b**).

**Figure 3 microorganisms-13-00921-f003:**
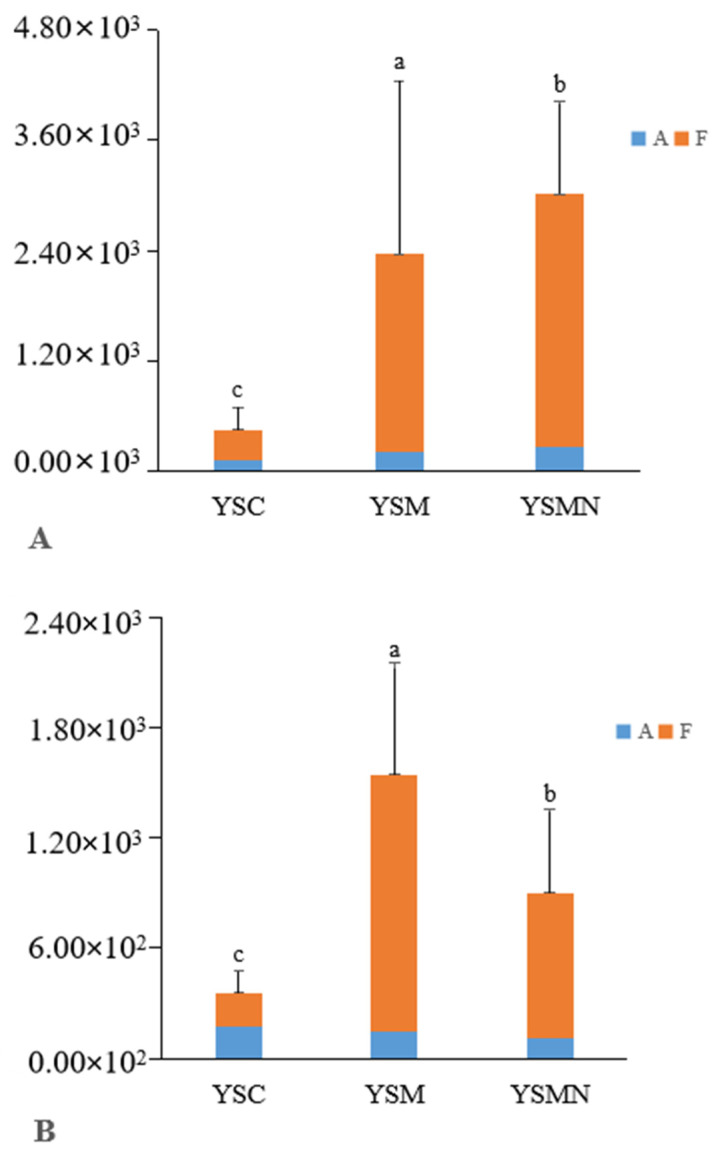
Effect of *Morchella* cultivation on bacteria copies(**A**) and fungi (**B**) copies in yellow sand substrate. A represents the time of sampling on day 0 after *Morchella* sowing. F represents the time of sampling on the 150th day after *Morchella* sowing. Different lowercase letters indicate significant differences (*p* < 0.05) in the different treatments.

**Figure 4 microorganisms-13-00921-f004:**
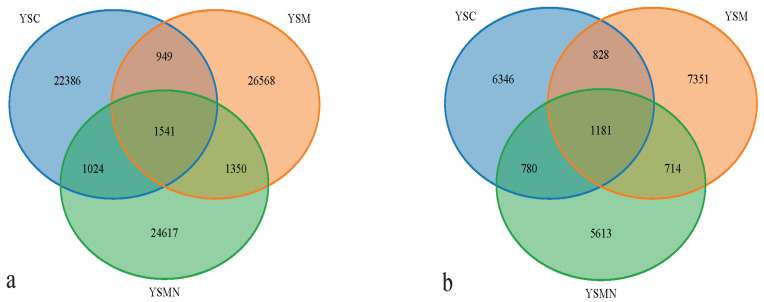
Effect of *Morchella* cultivation on the number of bacterial (**a**) and fungal microbial OTUs in yellow sand substrate (**b**).

**Figure 5 microorganisms-13-00921-f005:**
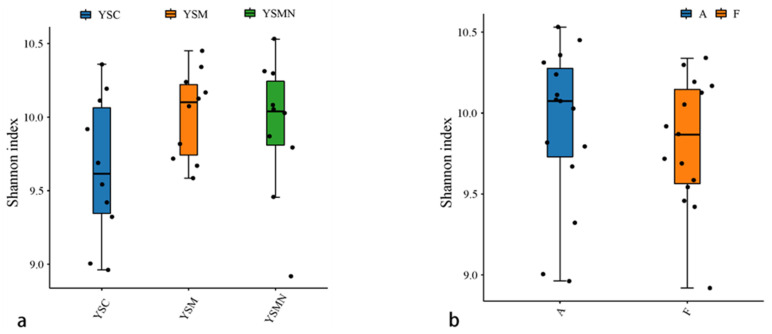
Microbial Alpha index of yellow sand substrate for *Morchella* Cultivation. (**a**) Bacterial in group; (**b**) Bacterial in time point; (**c**) Fungal in group; (**d**) Fungal in time point.

**Figure 6 microorganisms-13-00921-f006:**
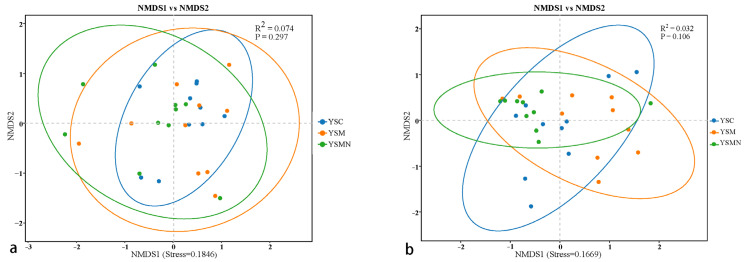
NMDS analysis of the bacterial community (**a**) and fungal community (**b**) in yellow sand substrate of *Morchella* cultivation.

**Figure 7 microorganisms-13-00921-f007:**
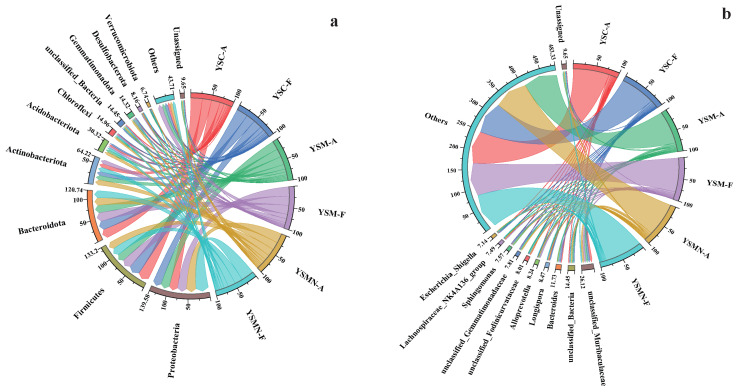
Taxonomic composition of the microbial communities. (**a**) Bacterial phylum; (**b**) Bacterial genus; (**c**) Fungal phylum; (**d**) Fungal genus.

**Figure 8 microorganisms-13-00921-f008:**
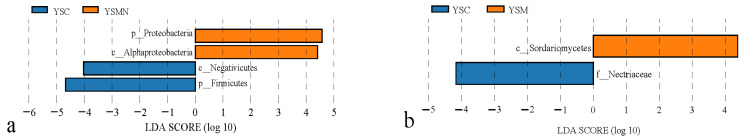
LEfSe analysis: Biomarker taxa of the bacterial communities (**a**) and fungal communities (**b**).

**Figure 9 microorganisms-13-00921-f009:**
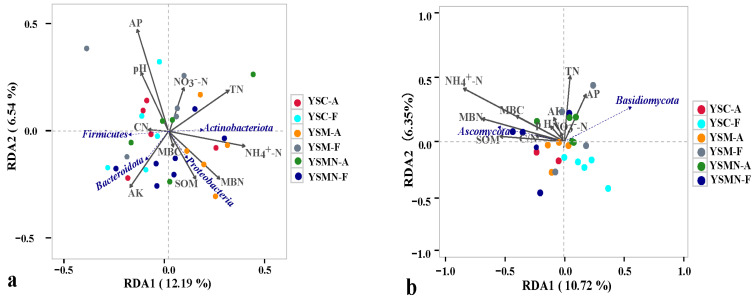
RDA analysis of the correlation between the bacteria community (**a**) and fungal community (**b**) and environmental factors in yellow sand under *Morchella* cultivation.

**Figure 10 microorganisms-13-00921-f010:**
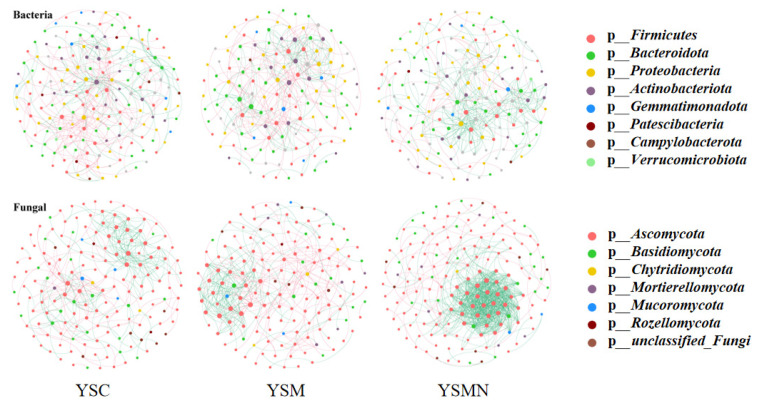
Co-occurrence network analysis of soil microorganisms under *Morchella* cultivation.

**Figure 11 microorganisms-13-00921-f011:**
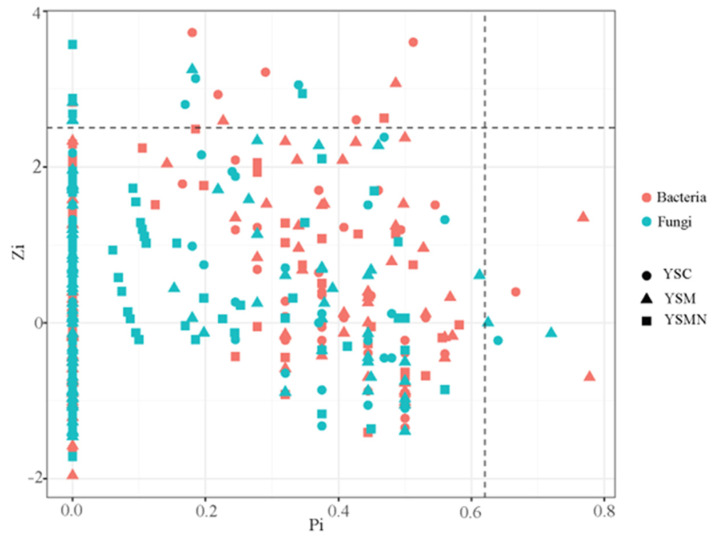
Topological roles analysis of soil microorganism co-occurrence network under *Morchella* cultivation.

**Table 1 microorganisms-13-00921-t001:** Main properties of the co-occurrence network.

Network Indexes	Bacteria			Fungal		
	YSC	YSM	YSMN	YSC	YSM	YSMN
Total nodes	171	138	163	164	152	165
Total links	345	308	316	352	333	678
Positive links	44.9	45.4	76.5	66.7	54.9	87.4
Average degree (avgK)	4.04	4.46	3.88	4.29	4.38	8.22
Average clustering coefficient (avgCC)	0.15	0.13	0.12	0.17	0.16	0.23
Average path distance (GD)	4.62	4.45	5.45	5.07	5.01	5.89
Transitivity (Trans)	0.16	0.16	0.19	0.26	0.34	0.68
Density (D)	0.02	0.03	0.02	0.03	0.03	0.05

**Table 2 microorganisms-13-00921-t002:** Contribution rates of deterministic processes and stochastic processes of yellow sand microbial communities under *Morchella* cultivation/%.

Category	Assembly Process	YSC	YSM	YSMN
Bacteria	Deterministic	65.44	68.89	95.56
	Stochastic	35.56	31.11	2.22
Fungal	Deterministic	20.00	20.00	51.11
	Stochastic	80.00	80.00	46.67

**Table 3 microorganisms-13-00921-t003:** Contribution rates of yellow sand microbial community assembly process under *Morchella* cultivation/%.

Category	Treatment	Homogeneous Selection	Heterogeneous Selection	Diffusion Limitation	Homogeneous Diffusion	Non-Dominant Process
Bacteria	YSC	17.78	46.67	4.44	17.78	13.33
	YSM	26.67	42.22	13.33		17.78
	YSMN		95.56			2.22
Fungal	YSC	11.11	8.89		55.56	24.44
	YSM		20.00		55.56	24.44
	YSMN	4.44	46.67		28.89	17.78

Blank indicates that the contribution of this assembly process is zero.

## Data Availability

The original contributions presented in this study are included in the article. Further inquiries can be directed to the corresponding author.

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
