# Peer review of "Microbial Community Response and Assembly Process of Yellow Sand Matrix in a Desert Marginal Zone Under Morchella Cultivation"

_microorganisms, 2025, doi:10.3390/microorganisms13040921_

Round 1

Reviewer 1 Report

Comments and Suggestions for Authors

The manuscript "Microbial community response and assembly process of yellow sand matrix in desert marginal zone under Morchella cultivation" present a interesting research approach on substrate-microbiome interaction.

The manuscript is based on multiple data and results, which are well organized and explored.

Some suggestions can be done to the current form of the manuscript in order to increase the presentation of the findings and a better explanation of the observed trends.

As a General comment, avoid very long sentences (e.g. Abstract section - lines 11-15). Try to split sentences that exceed 5 rows in two different one and to point one or two findings in each sentence. Please check the entire manuscript for these cases.

In the abstract include a short aim of this research. Split the first sentence and organize it - first one as the short aim, the second one as research objectives.

Introduction section - this part of the manuscript present the importance of Morchella cultivation, the current problems with yields and the necessity for more applied studies. The entire section present well the state of the art in the field. The aim and the objectives are presented at the end of the Introduction, in a long sentence. Lines 98-103 should be changed and rewritten. First sentence to present a clear aim, and another two or three sentences for the objectives proposed. 

Materials and Methods - this section offers all the necessary information to understand the experimental design, the techniques used in laboratory and the data analysis.

Results section - I suggest to make a multiple comparison test between variants, and add the letters to score the significance of differences. This will improve the presentation of the results (Figure 1.). The text associated with the interpretation is clear and extensive, all the results being explored.

Figure 2 - change the text in graph to english please, to be easy for readers to understand.

Figure 4 and 5, 6 - rewrite the figure caption to be understandable as stand-alone caption. 

For all figures caption add a legend with the abbreviation used in figures.

Expand the interpretation for figure 6. The NMDS ordination, even if it does not provide a clear differentiation between the three communities, show a different pattern of localization for more points in one treatment. You can focus on the multiple points that are i the same place, after you can interpret what are the differences in community assemblage that make a point to be far from the other ones in the same treatment. Use at maximum all the data from this figure.

The Results section is detailed and is based on numerous graphs.

Discussion section - the authors connect their study with other ones in the field and present the trends of their main findings.

Author Response

Point 1: As a General comment, avoid very long sentences (e.g. Abstract section - lines 11-15). Try to split sentences that exceed 5 rows in two different one and to point one or two findings in each sentence. Please check the entire manuscript for these cases.

Response 1: Thank you for pointing this out.We revised lines 11-15 of the abstract section, rewriting them as two sentences, and examined and revised long sentences throughout the paper,with a total of 20 long sentences revised.

Point 2: In the abstract include a short aim of this research. Split the first sentence and organize it - first one as the short aim, the second one as research objectives..

Response 2: We reorganized and revised the first sentence in the abstract according to the first short objective and the second research objective of reviewers' comments.

Point 3. The aim and the objectives are presented at the end of the Introduction, in a long sentence. Lines 98-103 should be changed and rewritten. First sentence to present a clear aim, and another two or three sentences for the objectives proposed.

Response 3: We have rewritten and revised the aim and the objectives of the study at the end of the introductionv according to the Reviewer's comments. We revised lines 98-103 along the lines of the first sentence specifying the objective and the other two indicating the proposed goal.

Point 4. Results section - I suggest to make a multiple comparison test between variants, and add the letters to score the significance of differences. This will improve the presentation of the results (Figure 1.).

Response 4: We performed multiple comparisons for all physical and chemical shape indicators in Figure. 1 and used lowercase letters to score the significance of differences. Different lowercase letters indicate significant differences (p < 0.05) in the same treatment under different sampling time-point,where black represents the YSC treatment, blue represents YSM treatment, and red represents YSMN treatment. 

Point 5. Figure 2 - change the text in graph to english please, to be easy for readers to understand.

Response 5: We apologize for the language problems in the original manuscript. We have changed the text of the horizontal coordinates in Figure 2. to English.

Point 6. Figure 4 and 5, 6 - rewrite the figure caption to be understandable as stand-alone caption.

Response 6: We have changed the graphic titles of Figures 4., Figures 5., and Figures 6. to stand-alone titles.

Point 7. For all figures caption add a legend with the abbreviation used in figures. 

Response 7: We have added legends to all of the figures and explained the legends and their abbreviations in the captions.

Point 8. Expand the interpretation for figure 6. The NMDS ordination, even if it does not provide a clear differentiation between the three communities, show a different pattern of localization for more points in one treatment. You can focus on the multiple points that are in the same place, after you can interpret what are the differences in community assemblage that make a point to be far from the other ones in the same treatment. Use at maximum all the data from this figure. 

Response 8: We are grateful for the suggestion. According with your advice, we extend the Figure 6. explanation to address the possible causes of sample point dispersion within the same treatment. The specific changes are on lines 359-374 and are in red font.

Reviewer 2 Report

Comments and Suggestions for Authors

The article is well-founded in the scientific literature consulted. In my view, the methodology is well described, and the results are adequately discussed and compared with the literature whenever possible. The conclusions accurately describe the research findings without overextending the information or speculating. In my opinion, the article is of sufficient quality to be published in its present form.

Author Response

The article is well-founded in the scientific literature consulted. In my view, the methodology is well described, and the results are adequately discussed and compared with the literature whenever possible. The conclusions accurately describe the research findings without overextending the information or speculating. In my opinion, the article is of sufficient quality to be published in its present form.

Response : We are very grateful to the reviewers for their comments and guidance on the manuscripts! We re-written some of the long sentences in the manuscript, while revising and refining minor issues.